# Monitoring Grassland Growth Based on Consistency-Corrected Remote Sensing Image

Yuejuan Ren [1,2], Qingke Wen [1,*], Fengjiang Xi [3], Xiaosan Ge [2], Yixin Yuan [1] and Bo Hu [1]

1 National Engineering Research Centerfor Geomatics (NCG), Aerospace Information Research Institute, Chinese Academy of Sciences, Beijing 100101, China
2 School of Surveying and Mapping and Land Information Engineering, Henan Polytechnic University, Jiaozuo 454003, China
3 Inner Mongolia Remote Sensing Center Co., Ltd., Huhhot 010000, China
* Correspondence: wenqk@aircas.ac.cn

**Abstract:** Monitoring grassland growth in large areas usually needs multiple images from different sensors or on different dates to cover the study area completely. Images from different sensors or on different dates need consistency correction to eliminate the sharp differences between images. The main contribution of this study is to promote a method for consistency correction of images on different days by constructing a linear regression equation of land cover types and the classification pixel mean. Taking a prefecture-level area in China as a test area, the consistency corrected images were applied for monitoring grassland growth. The results showed the following. First, compared with the normal correction equation constructed for two images, taking all features into account, the coefficient of determination of the equation corrected by the land cover types was improved, and the root mean square error was also significantly reduced. Secondly, the areas of consistency in the corrected image were improved compared with the original image, with an improvement rate of 21% for images from the same sensor and 25% for images from different sensors. The pixel average was much closer to the benchmark images, indicating that the corrected image was more consistent than the original image. Thirdly, when applied for monitoring grassland growth, consistency correction can solve the problem of misjudging grassland degradation. Grassland that was judged to be degraded using direct imagery, in fact, showed stable growth after consistency correction, and this type accounted for 7.33% of the regional grassland area. The seasonal characteristics of grass growth in the region were also obtained by monitoring the growth of grass in the region throughout the year. The application test showed that an effective image consistency correction method can improve the accuracy of grassland growth monitoring across a large area.

**Keywords:** monitoring grassland growth; consistency correction; linear regression equation; land cover types; classification pixel mean

## 1. Introduction

Monitoring grassland growth is a method of indirectly responding to grassland growth by processing remote sensing information at different periods based on the fact that remote sensing information has a close correlation with the condition of the grassland [1]. The application of remote sensing technology can permit continuous monitoring of and research into grassland, which is of great significance for the historical dynamics and status of the study area and for finding a sustainable development strategy for the surrounding area [2–5]. With the development of satellite remote sensing technology, more and more satellite sensors have been launched to provide a large amount of remote sensing data for monitoring grassland growth. However, the coverage of a large range area usually requires multiple images, which are affected by inconsistencies in the image acquisition time, cloud pixels, and atmospheric factors. These result in the problem of inconsistent

images within the range, which cannot reflect the spatial distribution characteristics of the range when monitoring grassland growth. Therefore, the correction of time consistency among non-same-day images is particularly important for monitoring grassland growth.

Remote sensing data from different days provide a large amount of information, and the use of time consistency can enhance the usefulness of data information and provide data sources for some practical application studies. The interactive comparison of remote sensing images can be based on both comparisons among the same series of satellites [6–10] and different series of satellites [11–15]. Interactive comparison of the spectral values of various remote sensing images can make the remote sensing data consistent and provide consistent remote sensing images for Earth observations across a large area. Therefore, many scholars have conducted related research. Among these studies, because the Landsat series satellite images have the longest running time, studies on the Landsat series satellite images themselves and interactive comparisons with other series satellite images have been constantly carried out. Mancino [16] conducted an interactive comparison study based on Landsat-7 ETM+ and Landsat-8 OLI images. She [17] conducted a comparative analysis of seasonal variation in vegetation based on Landsat series satellite images, while Li [18] used the average reflectance of vegetation indices in each band of the same image series to conduct a comparative analysis. The results showed that the reflectance values of each band of Landsat series satellite images had high consistency. Wu et al. [19] compared the apparent reflectance data based on two sets of GF-1, GF-2, and Landsat-8 images, and the analysis showed that the data in the same series of satellite images had high consistency, but there were some differences between them and Landsat-8 images. The analysis showed that images in the same series had stronger consistency than the non-same-series satellite images. However, the whole test area correction and sample mean correction used in the above studies and these methods can be affected by the spectral distortion of ground objects brought about by the unified correction of the overall ground objects. Therefore, because of the above shortcomings, it is very important to improve the correction method and improve the accuracy of the correction.

Medium and high-resolution images are advantageous for monitoring grassland growth. For example, Zhou et al. [20] used Sentinel-2 and Landsat-8 images to monitor grasslands in the central United States, and the results showed that the short return period of the Sentinel-2 images allowed them to build a complete time series model and provide more data for growth analysis. Garioud et al. [21] used optical and SAR images to monitor a grassland and showed more information about the grassland's properties. Meng et al. [22] used multi-source images for grassland monitoring and biomass estimation, and the results showed that high-resolution remote sensing images could improve the accuracy of grassland monitoring and biomass estimation. All of these studies indicated that the use of medium and high-resolution imagery can improve the accuracy and provide more data for grassland monitoring, so this study will prioritize the use of Sentinel-2 imagery for monitoring grassland growth. With the easy access and wide use of Sentinel images, its MSI sensor has a band range and band center closer to the Landsat OLI sensor, and some scholars have conducted cross-comparison studies between them. For example, Claverie et al. [23] studied the surface reflectance data products of both, and the chain derived three products that were processed for every Level-1 input product from Landsat 8-OLI and Sentinel-2 MSI, indicating that the reflectance values of the two are consistent. Hankui et al. [24] compared the two datasets and quantified their sensor differences by regression analysis, showing that the surface reflectance of MSI was greater than that of OLI on the same day and could be made consistent by regression. Robert et al. [25] evaluated the differences among sensor images on the same day using absolute difference metrics and long-axis linear regression in the continental United States, showing that the Sentinel MSI data are spectrally comparable with both types of Landsat image data. Cao et al. [26] harmonized the surface reflectance of Landsat-7 ETM+, Landsat-8 OLI, and Sentinel-2 MSI in China and used ordinary least squares linear regression to fit the equation. These studies are all cross-comparisons of Sentinel-2 MSI and Landsat-8 OLI data, indicating that the two

data sources have conditions for consistent correction, but these studies were all for the correction of same-day images, and the correction of non-same-day images will occur in practical applications. Therefore, correcting the consistency of images from different days is necessary for large-scale applications.

The medium and high-resolution images used in this study were directly acquired from GEE (Google Earth Engine) platform (https://code.earthengine.google.com/ accessed on 1 August 2022). GEE combines a multi-petabyte catalog of satellite imagery and geospatial datasets with planetary-scale analysis capabilities. The platform integrates common remote sensing image datasets including the Landsat series (Landsat) and medium and high-resolution imaging spectrometry (Sentinel) datasets over recent decades, covering climate, precipitation, night lighting, and land cover, in addition to Earth observation images [27,28], created with the rapid development of network technology and computer technology with cloud storage and cloud computing technology [29]. In the process of image pre-processing, this study overcame the limitations of traditional software processing by using a combination of the cloud-based and local spatial data processing platforms of GEE. Compared with traditional software processing, GEE uses its powerful backend server and has sufficient computing power to process and calculate remote sensing data [30], and there is almost no requirement for local hardware facilities. It is even possible to use GEE on a mobile phone for processing remote sensing images. This approach simplified many of the pre-processing processes of traditional remote sensing, thus greatly improving the efficiency and reducing the cost of processing remote sensing data [31–33].

In this study, we mainly used Sentinel-2 and Landsat-8 images and took Zhaotong City, Yunnan Province, as the research area. By distinguishing land cover types and performing regressions separately, the consistency of pixels on different dates is improved compared with direct image regression. Linear regression equations were sufficient for consistency correction, rather than other mathematical equations such as quadratic equations or sine functions. The main objectives of the study were the following. First, linear regression equations were constructed with land cover types to correct remote sensing images under different conditions on non-same-day. We constructed time-consistent images of Zhaotong City that are corrected to the same date for each area of the scope, making them comparable to each other. Second, the study aimed to monitor the growth of grassland in Zhaotong City based on the synthetic time-consistent images and compare them with the direct mosaic images to analyze the method's advantages in monitoring the growth of grasslands.

## 2. Data and Methods

### 2.1. Study Area

The study area was Zhaotong, Yunnan Province, China (Figure 1), which is located in the northeastern part of Yunnan Province on the Yunnan–Guizhou Plateau, at the junction of Yunnan, Sichuan, and Guizhou Provinces, at 26°55′–28°36′N and 102°52′–105°19′E. The topography is high in the southwest and low in the northeast, with an elevation of 267–4040 m, with a total area of 23,021 km$^2$ [34]. Due to its large extent and the influence of perennial cloudiness, each area within the range does not have consistency, making it more difficult to analyze the grassland growth in the study area. Therefore, it is of great importance to correct the time consistency of the non-same-day images of the study area.

### 2.2. Data Sources

#### 2.2.1. Remote Sensing Data

In this study, The Sentinel-2 MSI and Landsat-8 OLI sensor TOA (top-of-atmosphere) reflectance data were obtained through the GEE platform. The images obtained in this study have been geometrically and radiometrically corrected, and the original grayscale values of the images have been converted to apparent reflectance. All images that did not have a resolution of 10 m were resampled to 10 m using nearest-neighbor interpolation to maximize the preservation of the original spectral values due to the different spatial resolution of the data of the different sensors. Cloud pixels were identified and removed

from the image using the QA band, which was not involved in the subsequent processing and application. The coordinate system used for the image data was GCS_WGS_1984.

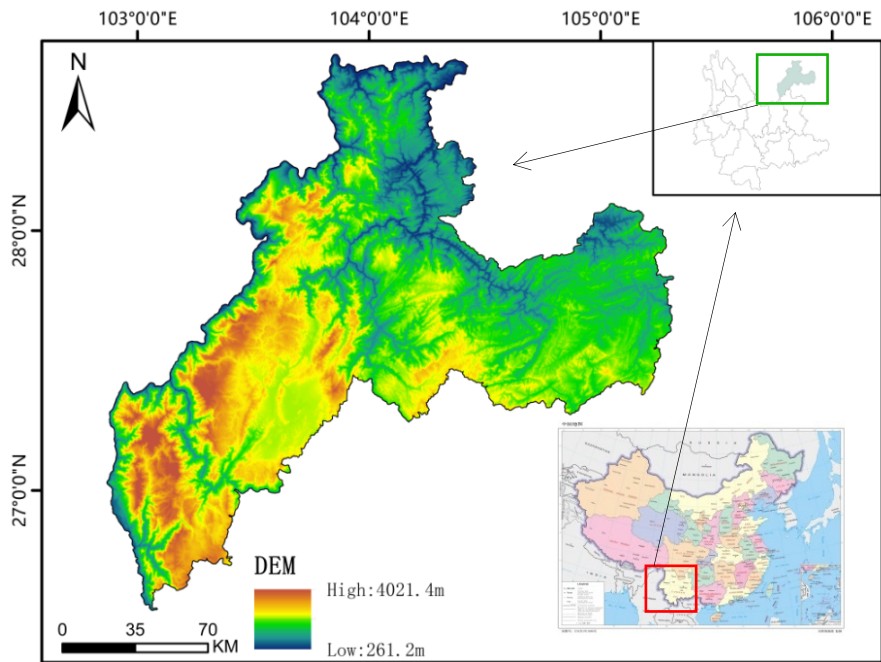

**Figure 1.** DEM data from the study area of Zhaotong City, Yunnan Province. The red frame shows the location of Yunnan Province in China, and the green frame shows the location of Zhaotong City in Yunnan Province.

Sentinel-2 MSI has 13 bands, and Landsat-8 OLI has only 9 bands. The application of this study was monitoring grassland growth, so only the NIR, red and green bands were used. In addition, the Sentinel-2 MSI NIR band is divided into the NIR wide band and the NIR narrow band. Since the NIR wide band is used in calculating the vegetation index and the spatial resolution of this band is also higher, the NIR wide band was used in this study for the Sentinel-2 MSI data. Table 1 shows the waveband information of the two sensors compared in this study.

**Table 1.** Details of the corresponding bands of Sentinel-2 MSI and Landsat-8 OLI used in this study.

| Sentinel-2 MSI | | | | Landsat-8 OLI | | | |
|---|---|---|---|---|---|---|---|
| Waveband | Spatial Resolution/m | Range/nm | Center/nm | Waveband | Spatial Resolution/m | Range/nm | Center/nm |
| B3 (Green) | 10 | 543–578 | 560 | B3 (Green) | 30 | 533–590 | 561 |
| B4 (Red) | 10 | 650–680 | 664 | B4 (Red) | 30 | 636–673 | 655 |
| B8 (NIR) | 10 | 785–900 | 843 | B5 (NIR) | 30 | 851–879 | 865 |

### 2.2.2. Land Type Thematic Data

The topography of Zhaotong City is complex and vulnerable to weather factors, and the accuracy of the classification results obtained by traditional methods of classifying land cover types is low. Therefore, this study used the land cover types classification results from the study by Yuan et al. [35], which mainly improved the input feature selection and sample selection, and then invoked the random forest algorithm in the GEE platform for classification. The final accuracy reached 88.21%. The geographical coordinate system used for the classification data was GCS_WGS_1984. The classification system used was cropland (10), forest (20), grassland (30), water (60), impervious surface (80), and bare land

(90), and their resolution was 10 m. Figure 2 shows the land cover types of Zhaotong City used in this study.

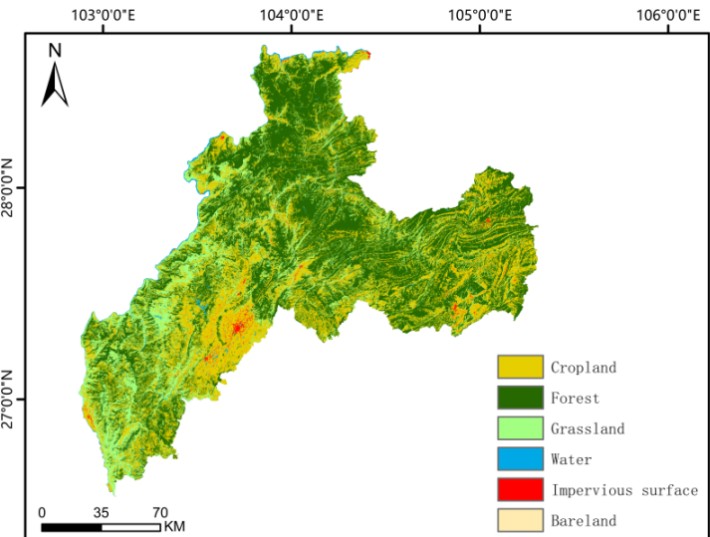

**Figure 2.** Zhaotong City's land cover types.

### 2.3. Research Methods

In this study, firstly, the corrected images comparable to the benchmark images were obtained by the consistency correction method. Secondly, the consistency images were constructed, and finally, the consistency images were applied to the grass growth monitoring in Zhaotong City. The technical flow chart of this study is shown in Figure 3.

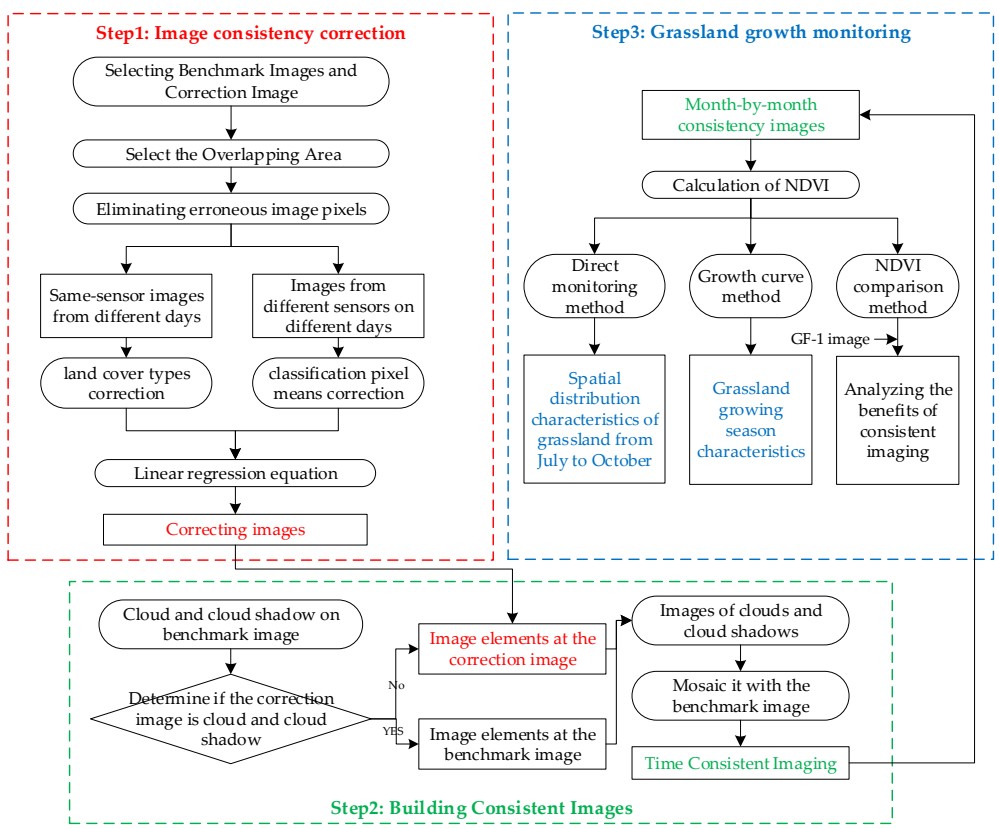

**Figure 3.** The technical flow chart of this paper.

### 2.3.1. Image Time Consistency Correction

(1)    Image and Overlap Area Selection

The complete coverage of the study area required nine same-day Sentinel-2 MSI images. Since the images are easily affected by clouds, shadows, and weather within the same day, resulting in missing data, the missing areas can be synthesized from the same sensor's images or those from different sensors (e.g., Landsat-8 OLI images), but this will cause the problem that the images of the missing areas in the study area will not have a time that is consistent with the original images. The Sentinel-2 MSI image from a certain date was defined as the benchmark image, and the image from the same sensor but a different date or the Landsat-8 OLI image were the images to be corrected to construct the time-consistent images of the study area from January to December 2021. The specific images used for each month are shown in Table 2. Figure 4 shows the distribution of the Sentinel-2 MSI and Landsat-8 OLI images. The selection of images is critical, where the benchmark image should be the one with the most complete coverage of the monthly image, and the image to be corrected should be one or more phases that can fill in the missing areas of the benchmark image and are dated close to the date of the benchmark image.

**Table 2.** The dates of the experimental images used to construct time-consistent images in this study.

| Month | Benchmark Image Sentinel-2 MSI | Image to be Corrected | |
| --- | --- | --- | --- |
| | | Sentinel-2 MSI | Landsat-8 OLI |
| 2021-1 | 01/01 | 01/14, 01/26 | - |
| 2021-2 | 02/10 | 02/20 | - |
| 2021-3 | 03/17 | 03/27 | |
| 2021-4 | 04/21 | 04/26 | 04/26 |
| 2021-5 | 05/01 | 05/21 | - |
| 2021-6 | 06/05 | - | - |
| 2021-7 | 07/20 | 07/25, 07/13, 07/30 | - |
| 2021-8 | 08/04 | 08/02 | - |
| 2021-9 | 09/23 | 09/21 | - |
| 2021-10 | 10/03 | 10/01, 10/18 | 10/01 |
| 2021-11 | 11/17 | 11/05 | 11/26 |
| 2021-12 | 12/22 | - | - |

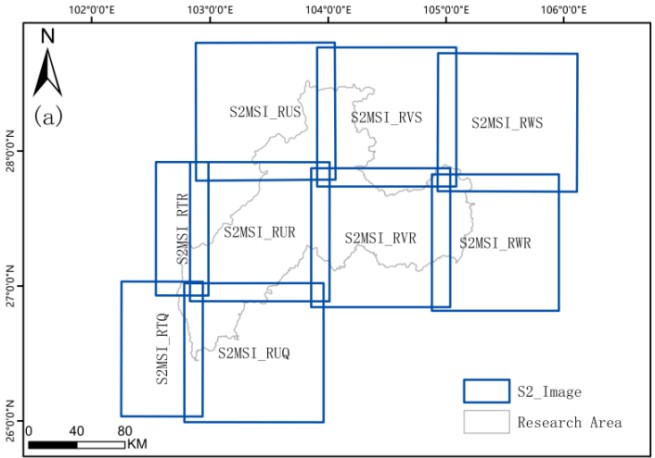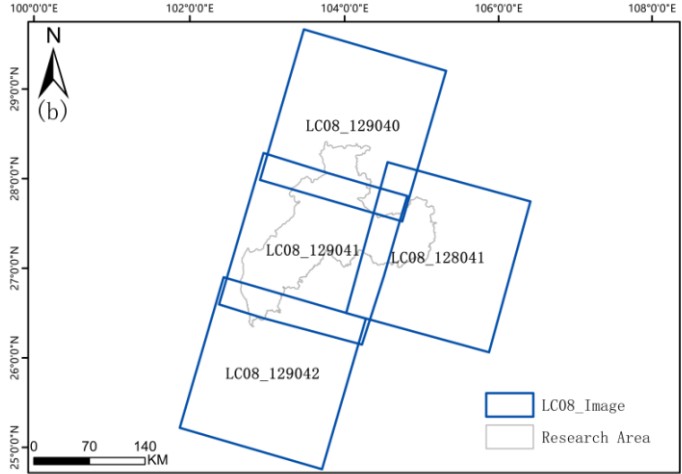

**Figure 4.** Image distribution required to fully cover Zhaotong City. (**a**) Sentinel-2 image distribution; (**b**) Landsat-8 OLI image distribution.

The sizes of the areas in the benchmark image and the image to be corrected are not the same, so to ensure the accuracy of the regression results, the largest overlap area of the two

images should be selected to establish the regression equation so that more image elements can be utilized. The left panel of Figure 5 shows the overlap area of the non-same-day Sentinel-2 MSI images, and the right panel shows the overlap area of the Sentinel-2 MSI and Landsat-8 OLI images.

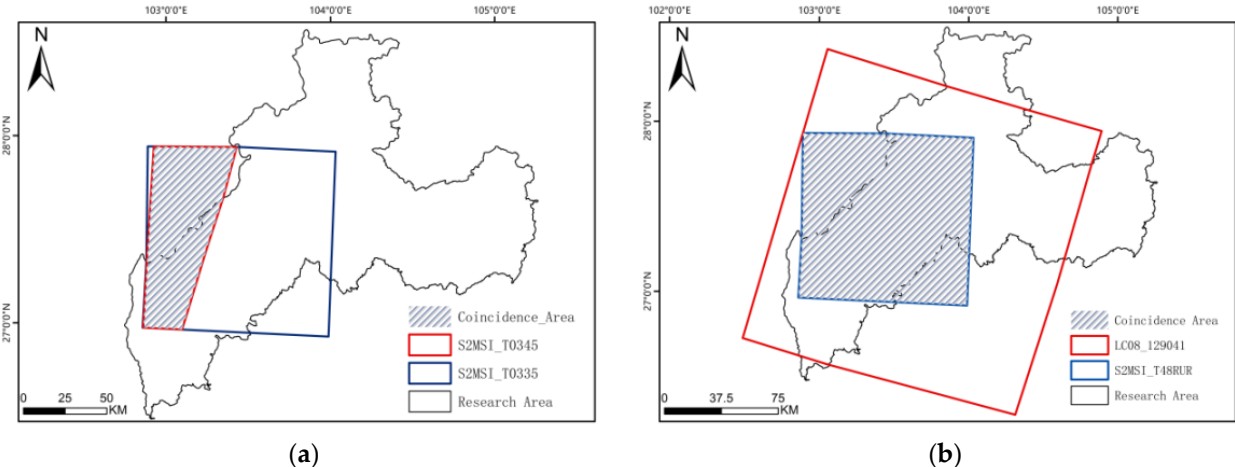

(**a**)  (**b**)

**Figure 5.** The overlapping area of the benchmark image and the image to be corrected in different situations. (**a**) Images from the same sensor (Sentinel-2) on different days; (**b**) Images from different sensors (Sentinel-2 MSI and Landsat-8 OLI) on different days.

(2)  Same-Sensor Images from Different Days: Land Cover Types Correction

The sample area averaging method [36] was used to select multiple areas of interest with the same extent in the overlapping area of the two images, and the mean apparent reflectance value of each area of interest was counted for the cross-comparison. This method effectively reduced the effects of inconsistent spatial resolution and alignment, but the selection of the areas of interest was affected by the inconsistency in the selected area, land cover types, and wavelength range, resulting in not all reflectance areas being selected, which made the cross-contrast accuracy low. Because of these shortcomings, this study proposed land cover types correction to improve them. Since the reflectance of different features in the same wavelength band is different (Figure 6), and the reflectance of each feature in the NIR band varies greatly, the feature classification results of Yuan [35] were used to establish the regression equations from the apparent reflectance value of each type of feature. This effectively avoided the problem of incomplete reflectance due to the selection of the region of interest and distortion of the feature spectrum caused by the uniform correction of all features and made the accuracy of the interaction comparison more accurate.

(3)  Images from Different Sensors on Different Days: Classification Pixel Means Correction

Since the resolution of both the classification data and the benchmark image is 10 m, and the original spectral values are maintained to the greatest extent possible, all images without a 10 m resolution were resampled to 10 m using the nearest neighbor interpolation method in this study. After we had increased the number of Landsat-8 OLI pixels, there were multiple apparent reflectance values from Sentinel-2 MSI corresponding to one identical apparent reflectance value from Landsat-8 OLI. Therefore, for time consistency corrections between different sensors, based on the land cover types correction method above and by establishing the regression equation with the mean value of the apparent reflectance for every 10 image pixels, the effect of duplicate image pixels on the time consistency correction of the images can be reduced to a certain extent. All pixels were selected from the overlapping area of two different sensor images, and the mean was obtained by traversing 10 pixels in turn for the pixels of the same land types. The order of pixel traversal was from the upper left corner to the lower right corner of the image of the overlapping

area. This provides methodological feasibility for cross-comparisons between images from different sensors with an inconsistent spatial resolution.

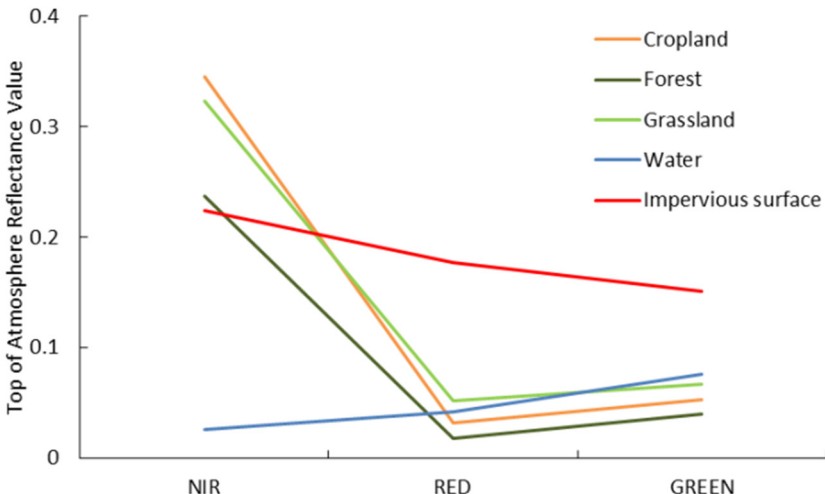

**Figure 6.** Spectral reflectance curves of different land cover types in the NIR, red and green bands.

(4) Eliminating Erroneous Image Pixels

The cloud image pixels were removed from the calculation in this study, but the image pixel under the cloud shadow still exists, and the apparent reflectance does not represent the true apparent reflectance value of the image pixel, and it is defined as an erroneous image pixel. A comparative analysis of the overlapping area between the benchmark image and the image to be corrected was performed, that is, the difference image of the two images was calculated. The image pixel under the cloud shadow causes the apparent reflectance value of the image to become smaller, and when this is subtracted from the other image pixel, the difference between the two becomes larger, i.e., the value of the difference image pixel becomes larger. Moreover, due to the influence of human activities and natural factors, the categories of two images on different days can change subtly, which can lead to deviations in the difference values. As shown in Figure 7, the red frame were pixels under the shadow of clouds, and their values were small. Therefore, the effects of cloud shadowing and the land cover classification results were combined, and the erroneous image pixels in the overlapping area of the two images needed to be removed before the cross-comparison. Firstly, all the image pixels in the different image were sorted in order of apparent reflectance value from smallest to largest, and the middle 80% of the raster data were converted into vector data, and most of the image pixel values outside the vector were the erroneous pixels values. Secondly, we used this vector data extraction method to obtain the new benchmark image and the image to be corrected. Finally, regression correction was performed on this basis to make the consistency results more accurate.

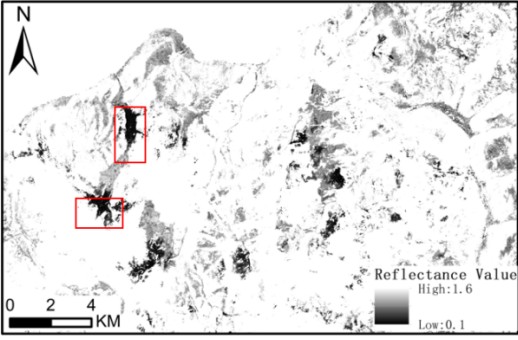

**Figure 7.** Apparent reflectance value of near-infrared images on a certain day, where the pixels in the red frame are under the shadow of clouds.

(5)    Accuracy Evaluation Index

The coefficient of determination ($R^2$), also known as the goodness of fit, characterizes the extent to which a regression equation explains the variation in the dependent variable or how well the equation fits the observed values. The greater the goodness of fit, the higher the degree to which the independent variable explains the dependent variable and the greater the percentage of the total variation caused by the independent variable. The root mean square error (RMSE) is the square of the deviation between the predicted and true values and the number of observations. In actual measurements, the number of observations n is always limited, and the true value can only be replaced by the most reliable (best) value, while the RMSE is used to measure the deviation between the observed value and the true value. In this study, these two evaluation indexes were used to measure the degree of deviation in different waveband interaction comparisons of images from the same sensor and those from different sensors. The equations used for this calculation are as follows:

$$R^2 = 1 - \frac{\sum_i \left( \hat{y}^{(i)} - y^{(i)} \right)^2}{\sum_i \left( \overline{y} - y^{(i)} \right)^2} \tag{1}$$

$$RMSE = \sqrt{\frac{1}{n} \sum_{i=1}^{n} \left( y^{(i)} - \hat{y}^{(i)} \right)^2} \tag{2}$$

where $y$ represents the true value, $\overline{y}$ represents the mean value, $\hat{y}$ represents the predicted value and n represents the total number.

### 2.3.2. Constructing Consistent Images

On the basis of the method above, the corresponding bands of the benchmark image and the image to be corrected for Zhaotong City (Table 2) were compared with each other to establish their regression equations. The regression equations were used to convert the image to be corrected into a corrected image that is time-consistent with the benchmark image. For clouds and cloud shadows in the benchmark image, if the corrected image was intact, the benchmark image was directly replaced, and if the corrected image also had cloud or cloud shadow, the benchmark image was kept. Figure 8 shows the process of synthesizing the benchmark image and the corrected image for April 2021. Firstly, the raster data of clouds and cloud shadows in the benchmark image and the correction image were converted to vector data. Next, the benchmark vector and the correction vector were processed with the erasure tool of ArcGIS to obtain the final vector data. Finally, the correction image was extracted with these vector data and synthesized with the benchmark image into a time-consistent image. Since there are only two data sources used in this study, and both of them have incomplete coverage during this month due to high cloud volume, there will also be incomplete images.

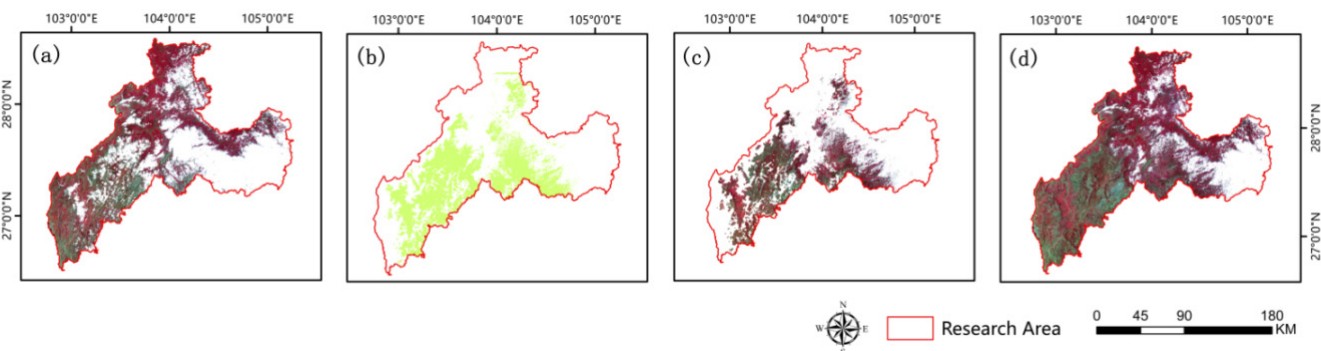

**Figure 8.** Time-consistent image composite for April 2021: (**a**) Benchmark image; (**b**) vector data; (**c**) correction image; (**d**) time-consistent image, synthesized from (**a**,**c**).

2.3.3. Monitoring Grassland Growth

Because of the large extent of the study area, the spatial distribution characteristics of the range could not be reflected in the direct mosaic images used for monitoring grassland growth, which led to misclassification of the results of monitoring grassland within the range. Therefore, in this study, the time-consistent images obtained by the above method are used to monitor the grassland growth and are subject to atmospheric correction and NDVI calculation before monitoring.

The main task of grassland growth monitoring is to reflect the growth status of grassland within the range. For the yearly grassland growth monitoring, the direct monitoring method is mainly used [37], and the direct monitoring method is to grade the vegetation indices obtained from remote sensing inversion so that the differences in the situation of grassland growth can be visually identified and determined. The vegetation growth process curve method can also be used [38,39]. The analysis of grassland growth condition should not only be done by direct growth monitoring but also by trend analysis and comparison from the time series curve. The vegetation growth process curve method is to construct the growth process curve based on the average value of NDVI obtained from remote sensing data at each time in the monitoring area, i.e., to draw the process curve of NDVI changes over time in the grass in the area, so as to analyze the growth situation of the grass based on the change characteristics of the curve. The growing season of grassland can be determined by using the distance level method [40], which is the difference between a value in a set of numbers and the average value and can be divided into positive and negative distance levels. The node of the NDVI curve value with a positive distance level is used as the threshold for the beginning of the growing season, and the node of the NDVI curve value with a negative distance level is used as the threshold for the end of the growing season.

## 3. Results and Analysis

### 3.1. Result of Eliminating Erroneous Pixels Correction

Taking the image in April 2021 as an example, the above method was used to eliminate erroneous pixels. Figure 9 shows the image comparison before and after eliminating erroneous pixels on a certain day, images a and b are the original image and the new image obtained after eliminating the erroneous image pixels, and the comparison shows that most of the reduced image pixels are erroneous ones. Regression analysis and comparison were carried out by using the apparent reflectance values of the NIR bands of the two pairs of images before and after elimination.

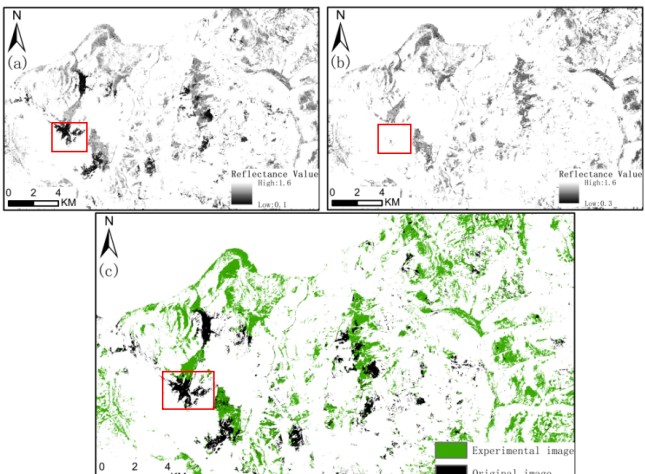

**Figure 9.** An example of the process of eliminating the erroneous pixels in a Sentinel-2 image. (**a**) Original image, where the pixels in the red frame are under the shadow of clouds; (**b**) experimental image, with no pixels in the red frame; (**c**) comparison image, where the pixels in the red frame are eliminated.

By comparing Figure 10 and Table 3, it can be seen that, compared with the correction of the image with erroneous pixels before the elimination, the $R^2$ values of the regression equation established by the correction after the eliminate are improved, the apparent reflectance scatter lines are all close to the 1:1 line, and the RMSE also decreased significantly. It shows that the correction after eliminating the erroneous pixels can improve the consistency between the images, which indirectly proves the necessity of eliminating the erroneous pixels. Figure 10 shows a comparison of the scatter distribution using the apparent reflectance of Sentinel-2 MSI on 26 April as the independent variable and the apparent reflectance of Sentinel-2 MSI on 21 April as the dependent variable, where N is the number of image pixels, the black dashed line is the 1:1 line and the red solid line is the regression line of the scatter plot.

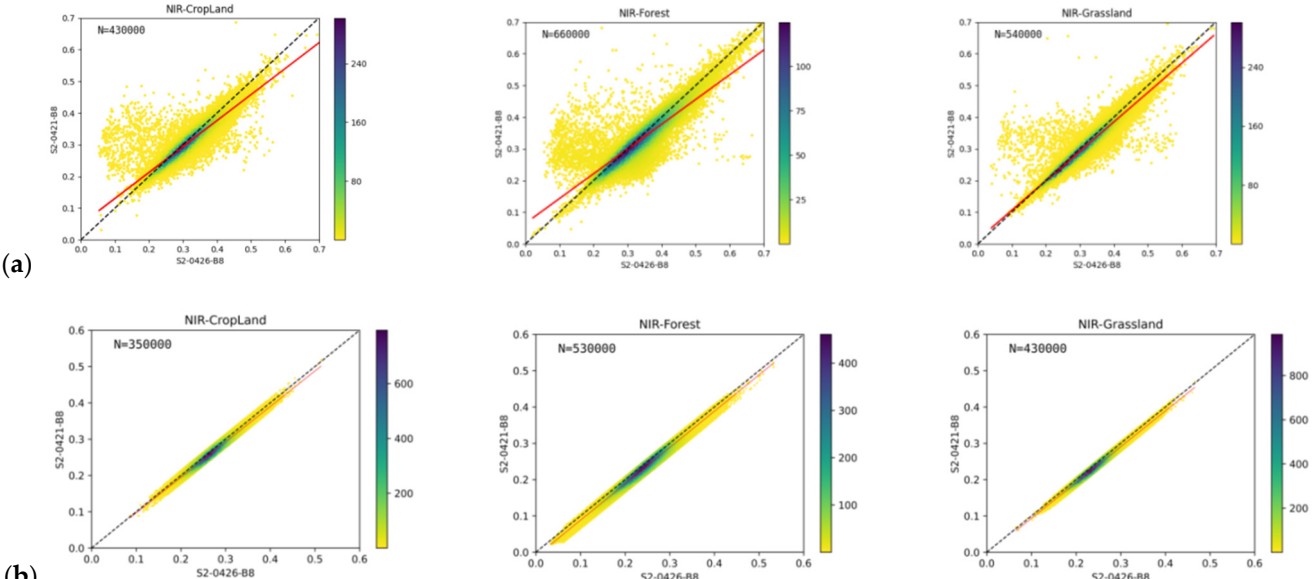

**Figure 10.** Comparison of apparent reflectance scatter points for correction before and after eliminating erroneous pixels. (**a**) Correction before eliminating; (**b**) correction after eliminating. N is the number of image pixels, the black dashed line is the 1:1 line, and the red solid line is the regression line of the scatter plot.

**Table 3.** Based on statistical comparisons between corrections before and after eliminating erroneous pixels.

| Category | Correction Method | Equation | $R^2$ | RMSE |
|---|---|---|---|---|
| Cropland | After eliminating | Y = 0.9769x − 0.0008 | 0.9361 | 0.0078 |
| | Before eliminating | Y = 0.8188x + 0.0486 | 0.8049 | 0.0224 |
| Forest | After eliminating | Y = 1.0005x − 0.0117 | 0.9558 | 0.0095 |
| | Before eliminating | Y = 0.7790x + 0.0659 | 0.7493 | 0.0402 |
| Grassland | After eliminating | Y = 0.9861x − 0.0052 | 0.9561 | 0.0061 |
| | Before eliminating | Y = 0.9285x + 0.0135 | 0.8943 | 0.0161 |

*3.2. Results of Non-Same-Day Image Correction*

3.2.1. Same Sensor Images Based on Land Cover Types Correction

In most correction comparison studies, the regression equation is affected by the overall feature correction and has spectral distortion. Therefore, in this study, we proposed land cover types correction to analyze the three features of cropland, forest, and grassland in the image. This approach makes the regression equations more specific, which leads to a more consistent image correction. The NIR bands of the Sentinel-2 MSI images of

the overlap area on 21 and 26 April 2021 were used to compare the apparent reflectance data of cropland, forest, grassland, and the whole overlap area by regression analysis. A comparison of Figure 11 and Table 4 indicates that the $R^2$ values of the regression equations established by the land cover types correction are all improved compared with the full test area correction, the apparent reflectance scatter lines are all close to the 1:1 line, and the RMSE is also significantly reduced. Figure 11 shows a comparison of the scatter distribution using the apparent reflectance of Sentinel-2 MSI on 26 April as the independent variable and the apparent reflectance of Sentinel-2 MSI on 21 April as the dependent variable.

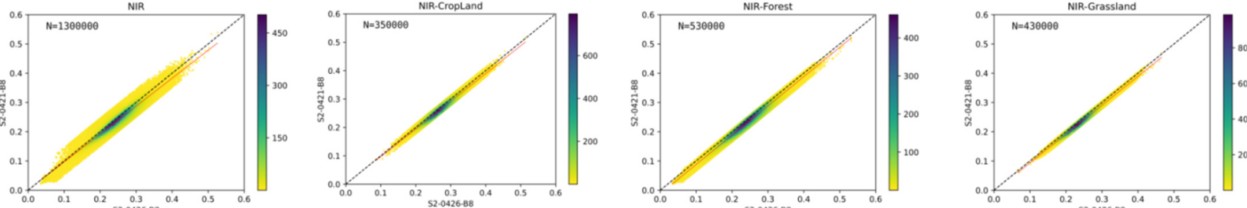

**Figure 11.** Comparison of the apparent reflectance scatter points for land cover types correction and whole test area correction. N is the number of image pixels, the black dashed line is the 1:1 line, and the red solid line is the regression line of the scatter plot. From left to right: Whole test area, cropland, forest, grassland.

**Table 4.** Statistical comparison between the land cover types correction and correction of the whole test area based on images from the same sensor.

| Different Correction Methods | Category | Equation | $R^2$ | RMSE |
|---|---|---|---|---|
| Whole test area | - | Y = 0.9568x − 0.0004 | 0.9005 | 0.0130 |
| Land cover types | Cropland | Y = 0.9769x − 0.0008 | 0.9361 | 0.0078 |
|  | Forest | Y = 1.0005x − 0.0117 | 0.9558 | 0.0095 |
|  | Grassland | Y = 0.9861x − 0.0052 | 0.9561 | 0.0061 |

The mean values of the image pixels of the three features in the corrected images obtained from the whole test area and the land cover types were calculated and compared with the mean values of the image pixels in the benchmark images (Table 5). It was found that the image averages of the land cover types corrected images were closer to the benchmark image's averages. Both methods could improve the consistency among the images, but the corrected images fit better with the benchmark images because land cover types correction is more targeted, which indirectly verifies the feasibility and necessity of land cover types correction.

**Table 5.** Comparison statistics of pixel means between the corrected images based on different methods and the benchmark image.

| Image | Correction Method | Cropland | Forest | Grassland |
|---|---|---|---|---|
| Benchmark image (04.21) | - | 0.2281 | 0.2272 | 0.2588 |
| Correction image (04.26) | Whole test area | 0.2259 | 0.2304 | 0.2527 |
|  | Land cover types | 0.2280 | 0.2296 | 0.2574 |

### 3.2.2. Different Sensors Images Based on Classification Pixel Means Correction

The regression analysis of the apparent reflectance data in the overlap region was performed using the NIR bands in the overlap region of the Sentinel-2 MSI images on 21 April 2021, and the Landsat-8 OLI images from 26 April 2021, with land cover types correction and classification pixel means correction. Figure 12 and Table 6 show that the $R^2$ values of the regression equations established by classification pixel means correction

improved by about 10%, and the RMSEs were reduced by about 40% compared with land cover types correction, and the scatter lines gradually converged to the 1:1 line. The analysis verified that classification pixel means correction can reduce the effect of duplicate image pixels on the time consistency correction of images from different sensors to some extent, and therefore, allows for a more accurate correction. Figure 12 shows a comparison of the scatter distribution using the Landsat-8 OLI apparent reflectance on 26 April as the independent variable and the Sentinel-2 MSI apparent reflectance on 21 April as the dependent variable.

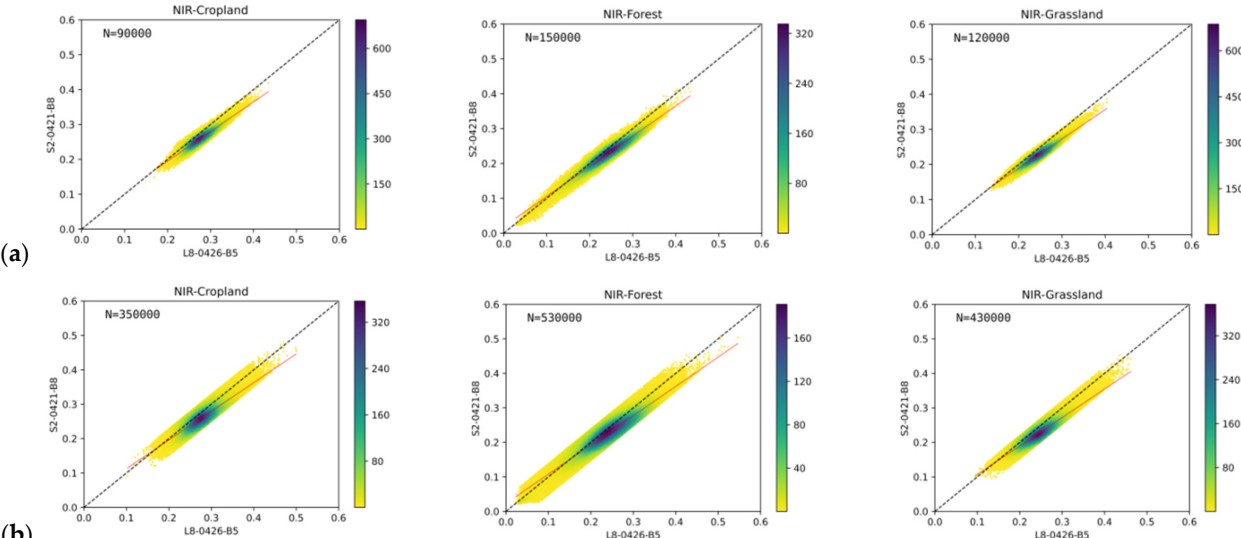

**Figure 12.** Comparison of apparent reflectance scatter points for land cover types correction and classification pixel means correction. (**a**) Classification pixel means correction; (**b**) land cover types correction. N is the number of image pixels, the black dashed line is the 1:1 line, and the red solid line is the regression line of the scatter plot.

**Table 6.** Statistical comparison between classification pixel means correction and land cover types correction based on the images from different sensors.

| Category | Correction Method | Equation | $R^2$ | RMSE |
|---|---|---|---|---|
| Cropland | Classification pixel means | Y = 0.8480x + 0.0253 | 0.8276 | 0.0097 |
| | Land cover types | Y = 0.8317x + 0.0299 | 0.7097 | 0.0165 |
| Forest | Classification pixel means | Y = 0.8647x + 0.0189 | 0.9108 | 0.0117 |
| | Land cover types | Y = 0.8468x − 0.0232 | 0.8352 | 0.0184 |
| Grassland | Classification pixel means | Y = 0.8381x + 0.0212 | 0.8717 | 0.0086 |
| | Land cover types | Y = 0.8232x + 0.0249 | 0.7808 | 0.0140 |

If we analyze and compare Tables 4 and 6, the correction of images from the same sensor is better than the correction of images from different sensors for different features, as shown by the improvement in $R^2$, the decrease in RMSE, and the more concentrated scatter. The spectral band centers and spectral ranges of the two sensors (Table 1 and Figure 13) indicate that the correction between images from different sensors has to overcome these factors, and therefore, the correction accuracy was lower. Moreover, the spatial resolution between the images from different sensors was different, and even though the use of classification pixel means correction can objectively smooth the effect and reduce the influence of duplicate images, it also makes the original apparent reflectance values deviate to a certain extent, while the cross-comparison between images from the same sensor was not influenced by these factors. Therefore, when performing time-of-month consistency

image corrections, images from the same sensor should be preferred to ensure the accuracy of the consistency correction.

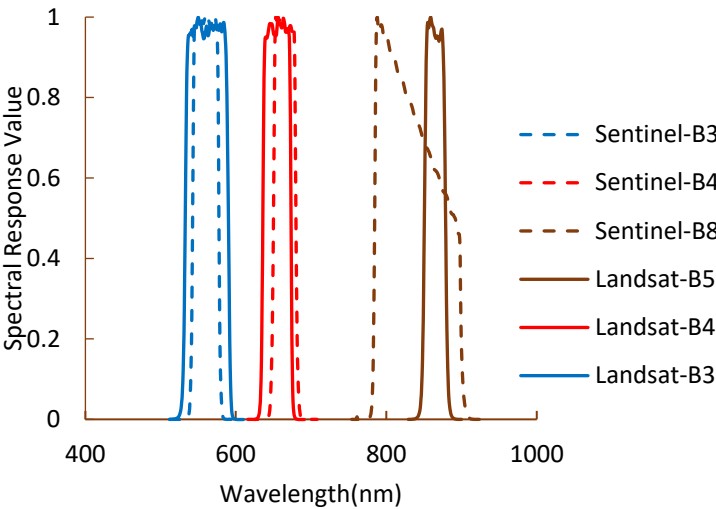

**Figure 13.** Spectral response function of Sentinel-2 MSI and Landsat-8 OLI.

*3.3. Comparison Results of Correction Images*

3.3.1. Comparison Based on Benchmark Images

To verify that the corrected image was more consistent with the benchmark image, experiments were designed using the overlapping area of images from the same sensor and the overlapping area of the images from different sensors (Figure 14) in April 2021. The corrected image of the overlapping area was obtained using the correction methods above. The difference between the NIR band apparent reflectance values of the benchmark image and the values of the image to be corrected and the corrected image were made, which were the true difference image and the corrected difference image, and $-0.02 \leq$ difference $\leq 0.02$ was defined as having consistency.

Through a statistical comparison, it was found that the area of consistency in the corrected difference images from the same sensor accounted for 82% of the total overlap area, while the area of consistency in the true difference images accounted for 61% of the total overlap area. The area of consistency in the corrected difference images from different sensors accounted for 52% of the total overlap area, while the area of consistency in the true difference images accounted for 27% of the total overlap area. This shows that the percentage of the consistent area increased for both images from the same sensor and images from different sensors. The mean values of all image pixels in the difference images were compared separately (Table 7), and the analysis showed that the mean values of the corrected difference images were closer to zero. In other words, the apparent reflectance values of the benchmark images and the corrected images were closer, indicating that the corrected images were more consistent with the benchmark images.

**Table 7.** Statistics of the pixel's mean values of the difference images from same sensor and from different sensors.

| Type | True Difference Image | Corrected Difference Image |
|---|---|---|
| Same Sensor | −0.0106 | 0.0019 |
| Different Sensors | −0.0221 | −0.0076 |

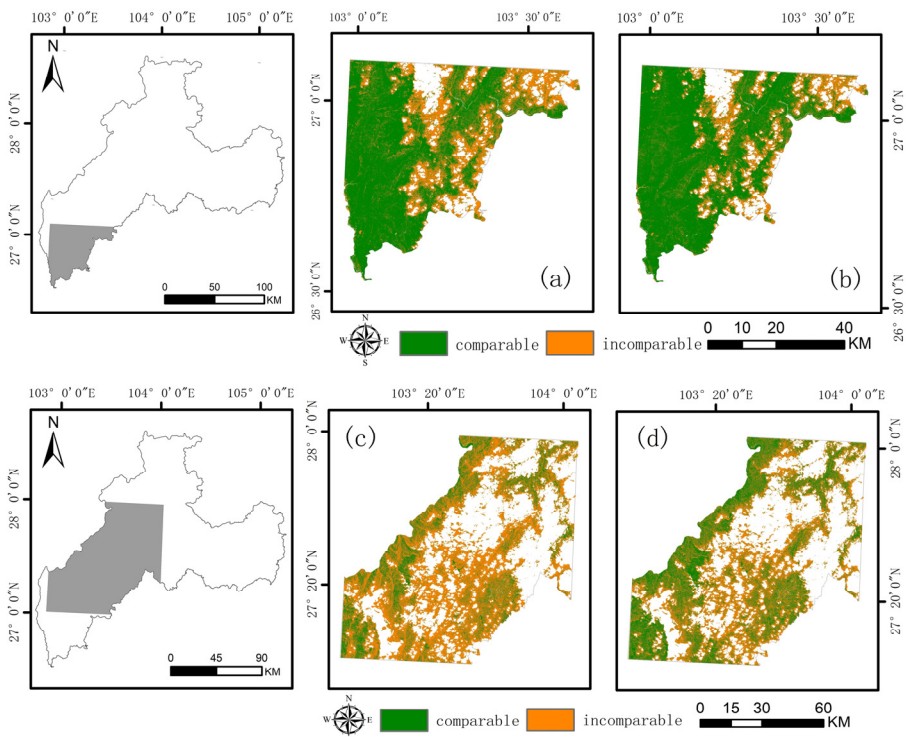

**Figure 14.** (**a**) True difference image under the same sensor, i.e., the difference between the pixels of the benchmark image and the image to be corrected; (**b**) Corrected difference image under the same sensor, i.e., the difference between the pixels of the benchmark image and the corrected image. (**c**) True difference image under different sensors; (**d**) Corrected difference image under different sensors. Pixels with $-0.02 \leq$ difference $\leq 0.02$ are defined as comparable; the rest of the pixels are incomparable.

### 3.3.2. Comparison Based on Synthetic Images

The corrected image and the image to be corrected were synthesized with the benchmark image to obtain a time-consistent image and a direct mosaic image (Figure 15). The junction of the mosaic images from different sensors and those from the same sensor images are shown in Figure 15a,b. By comparison, it can be seen that there are obvious changes in the direct mosaic images. In contrast, the time-consistent images did not show significant changes, and the fit was good, indicating that each area in the time-consistent image synthesized in this study was consistent.

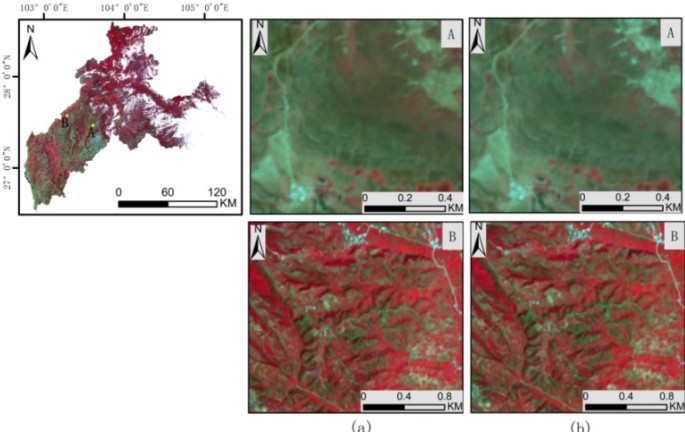

**Figure 15.** Synthetic image comparison: (**a**) Shows a column of direct mosaic images and (**b**) shows a column of time-consistent images. A is the junction of images from different sensors; B is the junction of images from the same sensor.

*3.4. Grassland Growth Monitoring Results*

3.4.1. Comparison of Grassland Growth Based on Different Images

To verify that the corrected images could more accurately reflect the spatial distribution characteristics of grassland growth, the NDVI comparison method was used to monitor the grassland growth in the direct mosaic image and the time-consistent image. A comparative analysis of grassland growth in April 2021 and 2022 was conducted, i.e., images of the difference in NDVI were calculated for both years. Using 2021 as the base year, growth was classified into three levels: Inferior, steady, and superior, corresponding to NDVI difference > 0.1, 0.1 ≥ NDVI difference ≥ −0.1, and NDVI difference < −0.1, respectively (Figure 16).

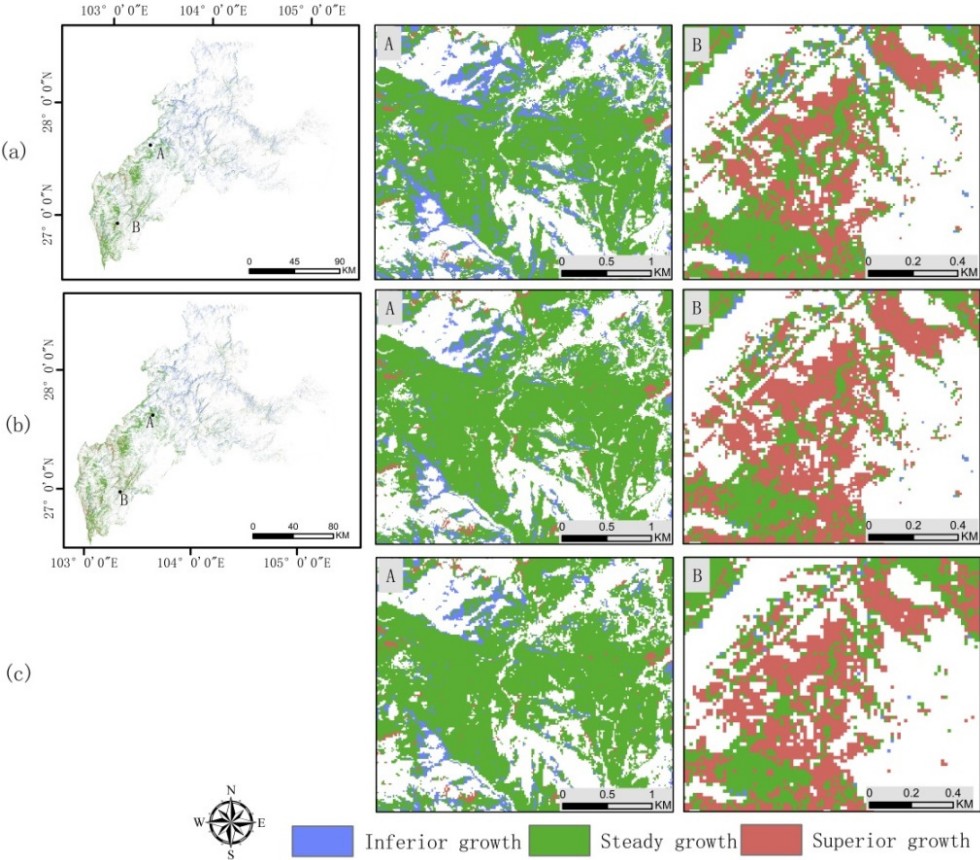

**Figure 16.** Comparison of grassland growth classifications from different synthetic images. (**a**) Is a row of direct mosaic images; (**b**) is a row of time-consistent images; (**c**) is a row of high-scoring images, which is used to verify the accuracy of the wrong classification of grassland growth in the method in this paper. Grassland growth in A was misclassified as inferior growth rather than steady growth. Grassland growth in B was misclassified as steady growth rather than superior growth.

According to the statistics in Table 8, it can be concluded that in the time-consistent image, the area of grassland in April 2022 that was inferior to that of the base year accounted for 31.76% of the total grassland area, the steady area accounted for 53.34%, and the superior area accounted for 14.90%. In the direct mosaic image, the area of grassland in April 2022 that was inferior to that of the base year accounted for 39.08% of the total grassland area, the steady area accounted for 50.41% and the superior area accounted for 10.51%. As shown in Figure 16a, compared with the time-consistent image, the grassland growth in the direct mosaic image was misclassified as inferior growth rather than steady growth, yielding 249.77 km$^2$ of grassland area subject to such misclassification, accounting for 7.33% of the total grassland area (Table 9). In contrast, the grassland growth in Figure 16b was misclassified as steady growth rather than superior growth, which yielded 149.77 km$^2$ of

grassland area subject to such misclassification, accounting for 4.39% of the total grassland area. This shows that the NDVI values on the direct mosaic images were inconsistent due to the inconsistent image acquisition time, which led to misclassification, while the time-consistent image synthesized in this study could more accurately reflect grassland growth across a wide range of days.

**Table 8.** Grassland growth area statistics based on different image synthesis methods (km$^2$).

| Image | Inferior Growth | Steady Growth | Superior Growth |
|---|---|---|---|
| Direct mosaic | 1332.03 | 1717.99 | 358.04 |
| Time-consistent | 1082.26 | 1817.99 | 507.81 |

**Table 9.** Statistics of grassland growth misclassification based on direct mosaic images.

| Misclassification Situation | Area/km$^2$ | Proportion of the Total Grassland Area |
|---|---|---|
| Steady growth → Inferior growth | 249.77 | 7.33% |
| Superior growth → Steady growth | 149.77 | 4.39% |

In order to test the accuracy of the misclassification of grassland growth, this paper uses the Gaofen-1 imagery in April 2021 and 2022. The data comes from China Remote Sensing Data Network (http://ids.ceode.ac.cn/ accessed on 5 September 2022.)and have undergone radiometric calibration, atmospheric correction, orthophoto correction, and NDVI calculations. The same NDVI comparison method was used to monitor grassland growth for two years. Since the purpose of the experiment is to verify the accuracy of the classification of the grass, only the experiments involving the misclassification of A and B are performed. In other places, there is a problem of incomplete image coverage due to the influence of cloud shadows.

Part c in Figure 16 was a grading diagram of grassland growth obtained through high-score images. Through comparison, the grassland growth in the high-scoring image at A was steady, and the grassland growth at B was superior. The growth of the grassland was the same as that in the time-consistent image. This indicates the accuracy of monitoring grassland growth on time-consistent images and the necessity of correcting for image time consistency.

3.4.2. Spatial and Temporal Characteristics of Grassland Growth

Using the method of constructing consistent remote sensing images to obtain the month-by-month consistent images of Zhaotong City in 2021, as shown in Figure 17, and because only two data sources (Sentinel and Landsat) are used in this study, and both of them make the monthly coverage incomplete due to more clouds, the coverage of consistent images from January to December 2021 is statistically 30.8%, 99.1%, 98.9%, 80.4%, 96.7%, 94.8%, 91.6%, 93.9%, 89.7%, 93.4%, 99.6%, and 68.4%, which is the maximum coverage that can be achieved by both Sentinel and Landsat data sources.

Based on the vegetation growth process curve method to reveal the law of the grassland growth process in Zhaotong City, the specific method is: Using the vegetation growth process curve method to calculate the average value of grassland NDVI on the consistent image month by month in 2021, and draw its grassland growth process curve according to the time series of each time period. The average value of NDVI from January to December to get the average value of NDVI in the whole year of 2021, and the difference between the monthly NDVI value and the NDVI value in the whole year to get the distance plot. The difference between the monthly NDVI values and the NDVI values for the whole year was obtained, and the distance plot is shown in Figure 18. The analysis shows that the growth cycle of Zhaotong grassland is from June to November each year, and the best month is August, after which the grassland starts to dry up gradually due to the influence of climate and rainfall, and the corresponding NDVI starts to decrease gradually.

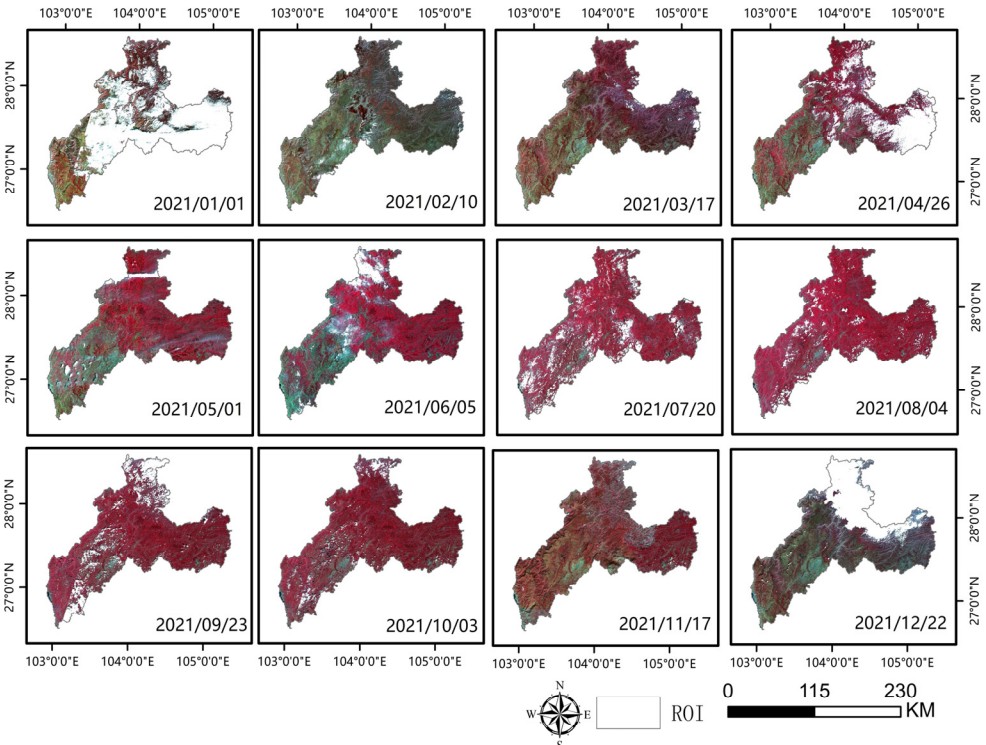

**Figure 17.** Time-consistent monthly images of Zhaotong City in 2021.

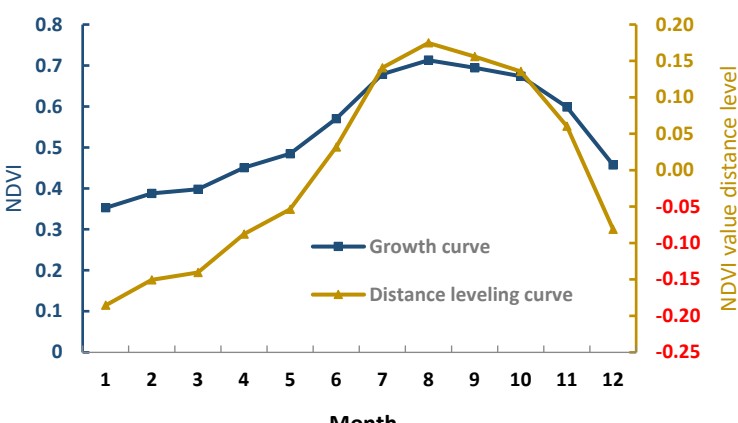

**Figure 18.** Grassland growth process curve and NDVI distance level in 2021.

The direct monitoring method was used to directly monitor the months of good grassland growth in Zhaotong City, mainly to reveal the differences in spatial distribution of grassland growth in the months of good grassland growth (July–October). The NDVI data for July–October 2021 in Zhaotong City were used to classify the growth classes at 0.1 intervals in order to monitor the differences in growth conditions within four months, as shown in Figure 19. Overall, compared to the monthly southwestern grasslands in Zhaotong, the northeastern grasslands generally grew better, with most of the northeastern grasslands having NDVI > 0.7 in four months, while there were significant differences in the NDVI of the southwestern grasslands, and the differences in growth in four months were also reflected in the southwestern part, with the southwestern grasslands growing worse in July and October, with most NDVI < 0.6, and better in August and September. The direct monitoring method can visually distinguish the spatial and temporal differences in grassland growth and can more easily analyze the growth of grassland in different regions.

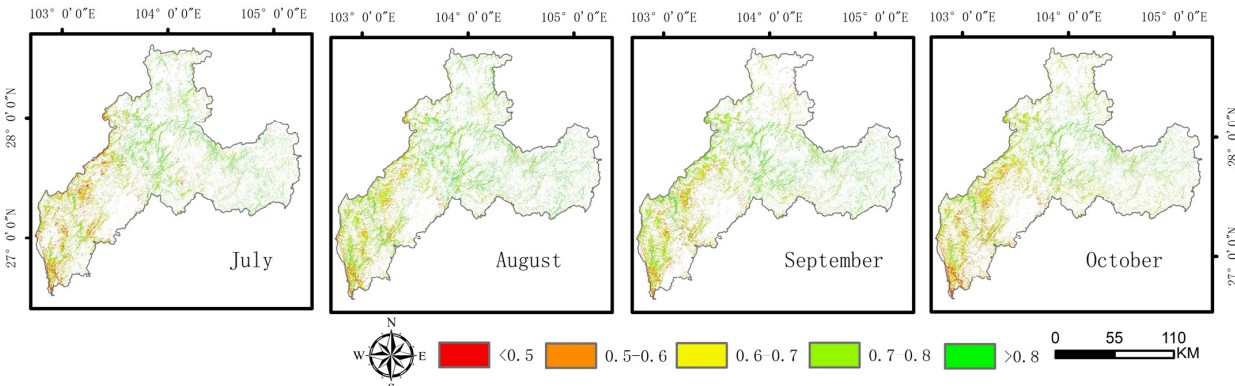

**Figure 19.** Classification of grassland growth from July to October 2021. Classify the NDVI values into 5 classes.

## 4. Discussion

Zhaotong City, Yunnan Province, is a large area, so the region requires multiple images for complete coverage. The images are susceptible to inconsistency in the acquisition time, and the images do not have consistency in different regions within the same time range. This limits the ability to monitor the growth of grassland in the area. Therefore, this study analyzed and investigated both the method of correcting the time consistency of the images and the use of time-consistent images for monitoring grassland growth.

As far as the time consistency correction method of the image is concerned, in this study, we have shown that non-same-day images from the same sensor can be corrected using land cover types, compared with the whole test area correction and sample mean correction used by previous researchers Robert [25] and Xu [36] et al. The method effectively avoided the effects of inconsistencies in the selected area, land cover types, and wavelength range, as well as the problem of distortion of the features' spectra due to uniform correction of all features, making the correction more accurate. For the correction between images from different sensors, we use classification pixel means correction, which can reduce the effect of repeated image pixels on the image time consistency correction to a certain extent. In addition, through experimental comparison, it is found that the accuracy of different sensors correction is lower than that of the same sensor. Therefore, other regression models can be used to construct different equations to correct for different sensors in future studies.

For monitoring grassland growth, we compared the direct mosaic image with the time-consistent image in this study. If the direct mosaic image was used for growth analysis, there were steep changes in the image, resulting in the misclassification of grassland growth. Using the corrected time-consistent images effectively avoided this problem and could be used to monitor the grassland growth more accurately. We used high-scoring imagery to validate the grassland growth classification and found that its growth situation in the same way as the time-consistent images. This study also applied the above method to construct consistent images of Zhaotong City for each month of 2021 and to monitor grassland growth throughout the year, demonstrating the transferability of the calibration method. Therefore, in future applications across wide areas, non-same-day images should not be used as a direct substitute but must be corrected to obtain time-consistent images over a larger range and to allow further analysis of their long time series. However, the time-consistent images obtained in this study still have incomplete coverage because only two data sources were used, and both of them had incomplete coverage due to a high cloud volume in the month studied. With the advancement of remote sensing technology, more and more data sources will become available for our use, so more data sources can be corrected in the future to make the time-consistent images more complete over a large area.

## 5. Conclusions

This study was based on Sentinel-2 MSI images of Zhaotong City on a certain day of the month as the benchmark images, correcting the Sentinel-2 MSI and Landsat-8 OLI images from the same month but on different days. Time-consistent images of Zhaotong City in monthly 2021 were obtained and monitored the growth of grassland. The rationality of the method used in this study was further analyzed. The major conclusions of this study are as follows:

(1) Different features have different reflectance. When a linear regression equation is constructed only by the reflectance values of the whole features for correction, the problem of spectral distortion of the features will be caused. This study constructed its own linear regression equations for different features through the land cover types and the classification pixel mean and applied them to image correction in different periods. It effectively solved the problem of low correction accuracy caused by the construction of linear regression equations for the whole features.

(2) Compared with the correction used for images from the same sensor, the accuracy of the correction used for images from different sensors was lower, which is likely to be caused by the different spectral range and spatial resolution. For spatial resolution, the use of classification pixel means correction can reduce its influence on the time consistency correction of images from different sensors to some extent, but the accuracy is still lower than that of the correction method for images from the same sensor. Therefore, same-sensor images should be preferred when constructing time-consistent images.

(3) Monitoring grassland growth in large areas usually needs multiple images from different sensors or on different dates, and the images need to be time consistent. This study promoted a consistency correction method, which effectively eliminated the sharp differences between large area images and improved the accuracy of grassland growth monitoring across a large area.

**Author Contributions:** All authors contributed in a substantial way to the manuscript. Y.R. and Q.W. conceived of, designed, and performed the research and wrote the manuscript. X.G., F.X., Y.Y. and B.H. contributed by revising the manuscript and examining the method. All authors have read and agreed to the published version of the manuscript.

**Funding:** This research was funded by the Inner Mongolia Autonomous Region Science and Technology Achievement Transformation Special Project (Grant No. 2020CG0123) and the Strategic Priority Research Program of Chinese Academy of Sciences (Grant No. XDA26050301-01).

**Data Availability Statement:** The data presented in this study are available on request from the corresponding author.

**Acknowledgments:** The authors would like to thank the Google Earth Engine platform for providing us with a free computing platform and free data. We are also thankful to all anonymous reviewers for their constructive comments provided on the study.

**Conflicts of Interest:** The authors declare no conflict of interest.

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
