# Peer review of "Monitoring Grassland Growth Based on Consistency-Corrected Remote Sensing Image"

_remotesensing, doi:10.3390/rs15082066_

Round 1

Reviewer 1 Report

Comments about the Paper: Monitoring Grassland Growth Based on Remote Sensing Image Consistency Correction Method”.

 The paper aims to promote a promote a method for consistency correction of images on different days. The results are promising especially for huge regions.

Nevertheless, there are some shortcomings that should be corrected. The methodological steps should be clearer (a flowchart is needed). How could the methods be transferred / generalized? Please provide a short description at discussion section.

In overall, the work is interesting and I believe the topic of this paper is appropriate to be published in Remote Sensing.

All in all, my recommendation is to accept the paper for publication, subject to minor revision. Please find in the attached file the amendments which I believe are required prior to accepting the paper.

Author Response

感谢老师的建议和指导,具体回复请参见附件。

Reviewer 2 Report

REVIEW SUMMARY

The authors present a manuscript to temporally align / correct (?) images from Sentinel-2 and Landsat 8 to make them better posed for agricultural grassland monitoring.

While the topic of agricultural grassland monitoring is an important one, I am not yet convinced of the approach. Since the manuscript is rather long, in places quite confusing, this could be one of the reasons.

My comments are as follows:

COMMENTS

[1] The title is a bit misleading because it suggests that the monitoring of grassland growth is based in the image consistency correction method. However, this is not the case. Instead, an NDVI-based method is used.

[2] It should be clearly defined what the authors understand as image consistency correction is and what the expected outcome is.

[3] Why not using (1) simply a best pixel selection (compositing), or (2) using an algorithm that does not require consistent images at all?

[4] The objectives are unclear. The first one is not measurable, the second one is “monitoring the growth of grassland”, which, by definition, involves a complete time series of the year. However, the authors present their study for images in April. This has two disadvantages: (1) It does not show the transferability of the approach since remote sensing is sensitive to climatic changes (e.g. variances of cloud cover throughout the year) and (2) this objective is not achieved with the study presented in the manuscript.

[5] The comparison that is presented in section 3.4 is in my opinion not sufficient to show monitoring of grassland growth. It may be that the authors and I have different assumptions of “grassland growth”; I would like to ask the authors to clarify what they mean.

[6] The entire section 2 would benefit from a Workflow figure, in particular since it is quite long. During reding, it was not very clear to me what the sequence of the individual steps are (and which will come). Some sub-sections are unclear in terms of why this step is applied.

[7] The conclusions are rather weak: The first one (use images from the same sensor has better performance than using images from different sensors) is somewhat trivial. Even more, I wonder why the authors did not consider using Sentinel-2A and Sentinel-2B? The second one dealing with monitoring of grassland can – in my opinion – to be concluded from the study for the reasons I explained above.

[8] Smaller comments

L88: “this study will prioritize the use of Sentinel-2 imagery for monitoring grassland growth” => Since Both Sentinel-2 and Landsat is used, this sentence is unclear to me.

L91: “MSI sensor data are closer to the Landsat OLI data” => closer compared to what?

L 158: Why not using Sentinel-2B as well?

L269: “all the image pixels in the different image (sic!) was sorted from the smallest to largest” => this is not clear to me

There are several changes of passive and active sentences “has been applied” vs. “we applied” => I suggest to make it consistent for better readability.

Some images are very difficult to read, e.g., Figure 8-11, due to the small font size.

A note on Figures 11 and 14: This is not good cartography because the red-green colour scheme is not readable by many people.

Author Response

Thanks teacher for your advice and guidance, please see the attachment for specific responses.

Reviewer 3 Report

see attached file

Author Response

(The authors gave the same response as above.)

Round 2

Reviewer 2 Report

The authors significantly improved the manuscript and performed several changes. Most of my comments were addressed; however, some still remain unclear, e.g., I still think that red-green colour schemes are not the best choice.